# Effects of Particulate Matter Inhalation during Exercise on Oxidative Stress and Mitochondrial Function in Mouse Skeletal Muscle

**DOI:** 10.3390/antiox13010113

**Published:** 2024-01-17

**Authors:** Jinhan Park, Junho Jang, Byunghun So, Kanggyu Lee, Dongjin Yeom, Ziyi Zhang, Woo Shik Shin, Chounghun Kang

**Affiliations:** 1Graduate School of Health and Exercise Science, Inha University, Incheon 22212, Republic of Korea; sportsjinhan@gmail.com (J.P.); jangju2489@gmail.com (J.J.); sportshun@inha.ac.kr (B.S.); skyarn@nate.com (K.L.); duarkans@gmail.com (D.Y.); 2Tianjin Key Laboratory of Exercise Physiology and Sports Medicine, Institute of Sport, Exercise & Health, Tianjin University of Sport, Tianjin 300381, China; zhangzy427@tjus.edu.cn; 3Department of Pharmaceutical Sciences, Northeast Ohio Medical University, Rootstown, OH 44272, USA; wshin@neomed.edu; 4Department of Physical Education, College of Education, Inha University, Incheon 22212, Republic of Korea

**Keywords:** particulate matter, oxidative stress, skeletal muscle, mitochondria, in vivo mitophagy, treadmill exercise

## Abstract

Particulate matter (PM) has deleterious consequences not only on the respiratory system but also on essential human organs, such as the heart, blood vessels, kidneys, and liver. However, the effects of PM inhalation on skeletal muscles have yet to be sufficiently elucidated. Female C57BL/6 or mt-Keima transgenic mice were randomly assigned to one of the following four groups: control (CON), PM exposure alone (PM), treadmill exercise (EX), or PM exposure and exercise (PME). Mice in the three-treatment group were subjected to treadmill running (20 m/min, 90 min/day for 1 week) and/or exposure to PM (100 μg/m^3^). The PM was found to exacerbate oxidative stress and inflammation, both at rest and during exercise, as assessed by the levels of proinflammatory cytokines, manganese-superoxide dismutase activity, and the glutathione/oxidized glutathione ratio. Furthermore, we detected significant increases in the levels of in vivo mitophagy, particularly in the PM group. Compared with the EX group, a significant reduction in the level of mitochondrial DNA was recorded in the PME group. Moreover, PM resulted in a reduction in cytochrome *c* oxidase activity and an increase in hydrogen peroxide generation. However, exposure to PM had no significant effect on mitochondrial respiration. Collectively, our findings in this study indicate that PM has adverse effects concerning both oxidative stress and inflammatory responses in skeletal muscle and mitochondria, both at rest and during exercise.

## 1. Introduction

Particulate matter (PM) is primarily derived from the combustion of coal, vehicle exhausts, and diverse industrial processes. It forms secondarily in the atmosphere through intricate chemical reactions involving sulfur dioxide and nitrogen oxides [1,2]. The size of PM is commonly classified as either coarse (PM_10_ < 10 µm) or fine (PM_2.5_ < 2.5 µm) [3]. When inhaled, suspended PM enters the body, and while coughing and saliva can remove some of the inhaled PM_10_ particles from the oral and nasal cavities, the remaining PM_2.5_ particles can readily penetrate the distal parts of the lungs, depositing in the alveoli [4,5,6]. As PM_2.5_ particles attach to alveolar tissues and can circulate in the bloodstream, acute or chronic exposure to PM can potentially contribute to an increase in systemic inflammation and oxidative stress [7,8,9]. Such exposure can accordingly exacerbate any preexisting respiratory, cardiovascular, and endocrine system disorders, including hypertension, chronic obstructive pulmonary disease, asthma, and type 2 diabetes [10,11,12].

Skeletal muscles play essential roles not only in movement but also as sites for a range of key biochemical processes, including the production of myokines, regulation of hormones, energy metabolism via the exchange of extracellular metabolites and oxygen, and intracellular signaling [13,14]. Evidence obtained to date indicates that similar to other organs, the detrimental effects of PM on skeletal muscle are associated with the circulation of proinflammatory cytokines stimulated by the presence of PM accumulating in the lungs. However, these discoveries are primarily based on the findings of in vitro studies [15,16,17], making it difficult to identify any specific systemic mechanisms. Accordingly, gaining more meaningful insights into the effects of oxidative stress and inflammatory responses provoked by PM deposition in the lung on skeletal muscle function and integrity, requires overcoming the limitations of in vitro studies by developing animal models that mimic atmospheric inhalation of particulate matter.

Prolonged aerobic exercise offers numerous health benefits, including enhancement of antioxidant and immune functions, which can contribute to inhibiting excessive inflammatory responses and the generation of reactive oxygen species (ROS), particularly in skeletal muscle [18,19,20]. Under normal physiological conditions, ROS play a key role in various cellular activities, including cellular energy metabolism, signal transduction, and the regulation of gene expression. However, when produced in excess, they can cause damage to cellular biomolecules, including lipids, proteins, and nucleic acids, thereby promoting cellular aging and eventually cell death [21]. High-intensity acute exercise, accompanied by a rapid increase in oxygen consumption, promotes excessive ROS production, leading to an imbalance in the oxidative–antioxidative homeostasis of cells [22]. However, in skeletal muscle adapted through long-term endurance training, the fibers undergo various structural adaptations and an increase in antioxidant enzyme activity, enhancing the capacity to adapt to high-intensity acute exercises and reducing oxidative stress levels [23,24]. The plasticity of skeletal muscle to adapt to exercise-related stress is characterized by an increase in myofibrillar cross-sectional area and changes in myofibrillar composition, promoted by resistance training and associated increases in metabolic and biochemical capacity. These adaptations are accompanied by morphological and functional changes, particularly in the mitochondria [25], the primary site of ROS production in response to exercise [26]. Morphological and functional changes in mitochondria, pivotal in cellular respiration and signal transduction, often correlate with elevated levels of oxidative stress in cells, and evidence indicates that PM may also influence mitochondrial homeostasis by inducing ROS production and inflammation [27].

The beneficial effects of exercise on the human body are widely acknowledged, but the repercussions of inhaling PM during physical activity are still being assessed, with a major focus on cardiorespiratory complications [28]. In this context, in the present study we sought to examine the effects of PM inhalation during exercise on oxidative stress, inflammatory responses, and mitochondrial function in the skeletal muscle of mice.

## 2. Materials and Methods

### 2.1. Experimental Animals

To eliminate the potential effects of testosterone on the accumulation of ROS and mitochondria-mediated cell death [29,30], female C57BL/6 mice (8 weeks old) were randomly divided into one of the following four groups: control (CON), PM exposure (PM), treadmill exercise (EX), and PM exposure + exercise (PME). For the study of in vivo mitophagy, mt-Keima transgenic (heterozygous type (+/−) FVB/N) mice (kindly provided by Dr. Jeanho Yun, Dong-A University) were bred and maintained in a specific pathogen-free facility. These mice harbor a mitochondria-targeting sequence derived from COX VIII that binds to the pH-dependent fluorescent protein Keima, thereby facilitating the detection of mitophagy. This model reflects the physiological status of mitochondria based on a dual-fluorescence probe, with a green fluorescence being indicative of normal conditions (pH 8.0) and red fluorescence signifying acidic lysosomal conditions (pH 4.5) [31,32]. All mice were housed in a temperature (22 °C)- and humidity (40–60%)-controlled environment illuminated on a 12 h light/12 h dark cycle, and had free access to allergen-free food and water. The protocols used in this study were approved by the Institutional Animal Care and Use Committee (IACUC: approval number INHA 190211-616, approval date 11 February 2019; TJUS: approval number TJUS 2022-021, approval date: February 2022).

### 2.2. PM Chamber and Treadmill Exercise

Commercial PM samples (Urban Particulate Matter; NIST1648A, Sigma, St. Louis, MO, USA) were used in a specifically constructed PM chamber designed to maintain a PM concentration of 100 μg/m^3^ during experimental assessments (Korea patent registration: 10-2529955 and 10-2529956, Republic of Korea). PM inhalation and/or exercise treatments were performed for 90 min per day for 7 days, with the incline and speed of the mouse treadmill being set at 20 m/min on a 5-degree uphill slope (Figure 1).

### 2.3. In Vivo Mitophagy Analysis

To evaluate in vivo mitophagy, we utilized mt-Keima transgenic mice. For in vivo observations, quadriceps muscle tissues obtained from mice were initially washed with cold phosphate-buffered saline, after which 1.0-μm-thick sections were cut using a brain slicer matrix and placed in confocal dishes (SPL). Nuclear staining was performed on ice using Hoechst 33342 and 4,6-diamidino-2-phenylindole solution (5 μg/mL) for 5 min (Thermo Fisher Scientific, Waltham, MA, USA). Fluorescence measurements were obtained using a laser confocal microscope (LSM 510 META; ZEISS) with excitation wavelengths of 488 nm (green) and 561 nm (red) and an emission wavelength of 620 nm. Changes in mitophagy were based on assessments of the red to green fluorescence ratio. Statistical analysis was conducted using ImageJ version 1.8.0 software.

### 2.4. Quantification of Mitochondrial DNA

The mitochondrial DNA (mtDNA) was purified from total DNA using a Nucleospin RNA Plus kit (MACHEREY-NAGEL, Düren, Nordrhein-Westfalen, Germany), with nuclear DNA (nDNA) being isolated using standard protocols. To quantify the amounts of mtDNA present per nuclear genome, we used the following primers pairs: mtDNA forward primer, 5′-CCTATCACCCTTGCCATCAT-3′ and mtDNA reverse primer, 5′-GAGGCTGTTGCTTGTGTGAC-3′; nuclear DNA forward primer, 5′-ATGGAAAGCCTGCCATCATG-3′ and nuclear DNA reverse primer, 5′-TCCTTGTTGTTCAGCATCAC-3′. The quantification of relative copy number differences was performed using the ΔΔCt method of the difference in threshold amplification between mtDNA and nuclear DNA. The RT-PCR thermal cycling conditions were 95 °C for 15 min and 50 °C for 40 s.

### 2.5. Enzyme-Linked Immunosorbent Assay

Interleukin-6 (IL-6), tumor necrosis factor-alpha (TNF-α), interleukin-1β (IL-1β), superoxide dismutase (SOD), and manganese superoxide dismutase (MnSOD) were measured using a Quantikine™ ELISA kit (R&D System, Inc., NE Minneapolis, MN, USA). For each assay, gastrocnemius muscle samples were prepared according to the manufacturer’s protocol.

### 2.6. Permeabilization of Muscle Fibers and Measurement of Respiration and H_2_O_2_ Generation

The permeabilization of muscle fibers and measurements of respiration and H_2_O_2_ production were performed using modified versions of previously described methods [33]. Samples of the red gastrocnemius muscle of mice (2–4 mg) were dissected and mechanically separated. The muscle fibers were permeabilized for 30 min using saponin (30 μg/mL) in buffer Z (pH 7.1; 30 mM KCl, 10 mM KH_2_PO_4_, 0.6 mg/mL BSA, 5 mM MgCl_2_-6H_2_O, 1 mM EGTA, 105 mM K-MES) supplemented with 1 mM EGTA (wash buffer), after which the preparations were washed three times with wash buffer. The permeabilized fiber bundles thus obtained were utilized to simultaneously measure the rates of oxygen consumption (OCR) and H_2_O_2_ generation using an Oroboros Oxygraph-2k device (O2k; OROBOROS Instruments, Innsbruck, Austria). OCR was assessed using an oxygen probe, while the rate of H_2_O_2_ production was evaluated using a green fluorescence sensor of the O2k-Fluo LED2 module. OCR measurements were standardized by incorporating antimycin A to account for non-mitochondrial oxygen consumption. The production of H_2_O_2_ was determined based on a standard H_2_O_2_ calibration curve.

### 2.7. Cyclooxygenase (COX) Activity Assay

COX activity was estimated using a COX activity assay (ab204699; Abcam, Cambridge, UK) according to the manufacturer’s instructions. Fluorescence (λEx/Em = 535/587 nm) was measured using a microplate reader in kinetic mode, and COX activity was expressed as μU/mg.

### 2.8. Malondialdehyde (MDA) Measurements

For the determination of the levels of MDA in gastrocnemius muscle tissues, 20 mg samples were initially homogenized in 250 μL of 7.5% trichloroacetic acid. After centrifugation and filtration, the resulting supernatants were combined with an equal volume of a mixture containing 10% trichloroacetic acid and 0.5% TBA. The samples were then boiled in a dry thermoblock for 30 min, followed by cooling. The absorbance of the TBA–MDA complex thus obtained was measured at 532 nm and corrected for non-specific absorbance at 600 nm to account for background noise.

### 2.9. Glutathione/Oxidized Glutathione Assay

Total and oxidized levels of glutathione (GSH and GSSG, respectively) were determined using a Glutathione Assay Kit provided by Cayman Chemical Company. Assays were performed by initially homogenizing gastrocnemius muscles in cold buffer (50 mM MES, pH 6 to 7, 1 mM EDTA) followed by centrifugation at 10,000× *g* for 15 min at 4 °C. The resulting supernatants were deproteinized using MPA reagent, followed by further centrifugation at 3000× *g* for 2 min. The resulting supernatants were treated with TEAM Reagent to quantify the total GSH present. To determine the levels of GSSG, we added 2-vinylpyridine to the supernatants and determined the level of reduced GSH by subtracting GSSG from the total GSH content.

### 2.10. Statistical Analysis

Data were analyzed using SPSS 22.0 software for statistics and GraphPad Prism 8 (v8.0.2, 2019) for visualization. The normality of the distribution for outcomes was assessed using the Shapiro–Wilk test and QQ plot. The differences between groups were tested via the student’s t-test, by comparing the groups. The interaction between PM exposure and treadmill exercise was analyzed via a two-way analysis of variance (ANOVA). Test were two-tailed, and significance was set at *p* < 0.05. All values are presented as the means and standard error (SEM).

## 3. Results

To assess the generation of lipid peroxidation by-products in gastrocnemius muscles, we measured the levels of MDA. Compared with the CON and EX groups, we detected a significant increase in MDA levels in the two respective PM inhalation groups (*p* < 0.001; Figure 2A). To assess the levels of proinflammatory cytokines, we measured the production of IL-6, TNF-α, and IL-1β. Compared with the levels in mice in the CON group, the levels of IL-6 were found to be 2.75-fold higher in the PM-treated mice (*p* < 0.001; Figure 2B). Similarly, compared with the levels in the EX group, the levels of IL-6 were found to be 49% higher in the PME group mice (*p* < 0.001). Likewise, compared with the levels in the CON and EX groups, we detected 2.6- and 4.0-fold higher levels of TNF-α in the PM and PME groups, respectively (*p* < 0.001; Figure 2C). Notably, two-way ANOVA revealed a significant interaction between exercise and PM inhalation regarding TNF-α levels (*p* < 0.05). Furthermore, we detected increases of 79% and 90% in IL-1β in response to the PM and PME treatments, respectively (both *p* < 0.001; Figure 2D).

We detected no significant differences among groups regarding total SOD levels. The levels of MnSOD, the only SOD enzyme located within the mitochondrial matrix, were found to be 30% lower in the PM group compared with that in the CON group (*p* < 0.01; Figure 3B) and 40% lower in the PME group compared with that in the EX group (*p* < 0.01). We also observed a significant reduction of 7.3% in the total GSH content in response to PM treatment (*p* < 0.05), and a reduction of 17% in PME mice compared with that in the EX group (*p* < 0.01; Figure 3C). Conversely, we recorded a 29% increase in the levels of oxidized GSH (GSSG) in the PM group compared with that in the CON group (*p* < 0.05), and an increase of 19% in the PME group compared with that in the EX group (*p* < 0.05; Figure 3D). Moreover, compared with the CON and EX groups, we detected reductions of 25% and 28% in the ratio of GSH to GSSG (GSH/GSSG) in the respective groups exposed to PM (*p* < 0.01; Figure 3E).

Further analysis of the levels of in vivo mitophagy, using model transgenic mice that express the pH-dependent fluorescent protein mt-Keima, revealed a significant increase in the levels of mitophagy (as indicated by an increase in the red/green fluorescence ratio) in the PM, EX, and PME groups compared with that in the CON group (*p* < 0.001). Interestingly, although we detected a significant increase in mitophagy in mice subjected to exercise (*p* < 0.001) compared with the CON group mice, this increase was significantly reversed following exposure to PM (*p* < 0.01; Figure 4A). To examine mitochondrial biogenesis, we assessed the ratio of mtDNA to nDNA and, in line with expectations, detected a significant exercise-induced increase in this ratio in the CON and PM group mice (*p* < 0.001; Figure 4B). However, compared with the CON or EX groups, we detected no significant difference in those mice exposed to PM.

To evaluate mitochondrial function, we initially analyzed the activity of mitochondrial cytochrome *c* oxidase (COX), which plays an essential role in ATP production, and accordingly detected reductions in activity of 8.5% and 32% in PM vs. CON (*p* < 0.01), and PME vs. EX (*p* < 0.001) comparisons, respectively (Figure 5A). Moreover, we detected an interactive effect between PM inhalation and exercise (*p* < 0.05). We then evaluated mitochondrial respiration and H_2_O_2_ generation in permeabilized red gastrocnemius muscle fiber bundles, and in line with expectations, recorded an exercised-induced increase in the rate of mitochondrial complex I + II oxygen consumption (+glutamate, malate, ADP, and succinate) (*p* < 0.001; Figure 5B). In contrast to COX activity, compared with the CON and EX groups, we detected no significant changes in the corresponding groups in which mice had been exposed to PM, although non-significant reductions in levels were detected in the PME group compared with those in the EX group. Similarly, there were no significant exercise-associated differences between the CON, PM, EX, and PME groups concerning respiratory control ratio values (*p* < 0.001; Figure 5C). However, exercise was found to promote a significant increase in the generation of state I H_2_O_2_ (*p* < 0.001), with exposure PM also inducing the production of 48% and 60% higher levels of H_2_O_2_ compared with those of CON and EX groups, respectively (*p* < 0.001; Figure 5D).

## 4. Discussion

It is assumed that PM has adverse effects on skeletal muscle integrity, either through direct deposition in skeletal muscle tissues or by inducing the release of proinflammatory cytokines in the lungs. In this study, we accordingly examined the effects of PM inhalation on the redox status, inflammatory responses, and mitochondrial function of skeletal muscle. To date, experimental animals have been injected with or exposed to different concentrations of PM to gain insights into the mechanisms by which these particles influence the function and integrity of different body tissues [34]. In the present study, we developed a novel experimental model based on a PM generation chamber containing a mouse treadmill designed to mimic actual human PM inhalation conditions (Figure 1), using which, we examined the effects of PM inhalation during exercise on selected dependent variables. The findings of several studies conducted to date have indicated that exposure to PM increases oxidative stress in the pulmonary and cardiovascular systems, leading to lipid peroxidation and subsequent downstream inflammatory signals [9,35]. It has also been demonstrated that in addition to the lungs, deposits of PM can accumulate in a number of other organs, including the brain, liver, heart, and peripheral blood vessels [13,36]. These findings accordingly indicate that by triggering local and/or systemic inflammation and oxidative stress, prolonged exposure to PM may heighten the risk of direct damage to multiple vital organs [37]. In the present study, we also detected elevated levels of lipid oxidation (as evaluated by the oxidative stress-related production of MDA) in the gastrocnemius muscle of mice exposed to PM, signifying that PM may also have a detrimental effect on skeletal muscles.

Biological ROS and reactive nitrogen species can be generated in different cellular compartments, including the mitochondria, peroxisomes, endoplasmic reticulum, and phagocytes. An excessive production of ROS can disrupt cellular homeostasis and compromise the immune system [38], and multiple studies have provided evidence of elevated levels of oxidative stress in skeletal muscle following exercise, associated with the production of diverse ROS free radicals, including superoxide anion radicals (O_2_^•−^), hydroxyl radicals (·OH), hydroperoxyl radicals (HOO.), singlet oxygen (1O_2_), and free nitrogen radicals [39]. Consistently, in the present study we established the inhalation of PM exacerbates oxidative stress, as evidenced by a reduction in MnSOD activity and GSH/GSSG levels. However, heightened oxidative stress in skeletal muscles, resulting from either exercise or PM exposure, can engender varying degrees of adaptability. For example, an acute spike in exercise-induced oxidative stress is unlikely to culminate in chronic problems. However, due to the establishment of higher antioxidant capacities promoted by regular exercise [40,41], this may not hold true for PM exposure.

Concerning cytokine production, a previous study on humans, in which the authors measured the levels of IL-6 and TNF-α in the exhaled breath condensate of individuals with and without asthma, revealed a strong correlation between the concentration of PM and elevated levels of proinflammatory cytokines in people with asthma [42]. Consistently, other studies have established a positive association between short-term exposure to PM_10_ and elevated levels of circulating IL-1β, IL-6, and TNF-α in the general adult population. These positive correlations accordingly indicate an association between air pollution and heightened cardiovascular risk [43]. Furthermore, the findings of other studies have revealed elevated levels of proinflammatory cytokines in different organs, including the brain, kidneys, and lungs, of animals exposed to different concentrations of PM, either via inhalation or injection [37,44,45,46]. In the present study, we similarly found that respiratory inhalation of PM during exercise induced an increase in proinflammatory cytokines in the skeletal muscles of mice. Rapid fluctuations in the levels of IL-6, IL-1β, and TNF-α within skeletal muscle imply their role in regulating muscle cell degradation and apoptosis, as well as muscle fiber atrophy and hypertrophy [47]. These responses would thus tend to indicate that prolonged exposure to heightened levels of proinflammatory cytokines is associated with unfavorable physiological outcomes [48]. Notably, we detected a correlation between PM inhalation and exercise regarding their effects on TNF-α, indicating the necessity for further mechanistic studies to gain a more comprehensive understanding of the specific mechanisms by which PM inhalation stimulates the release of TNF-α during exercise. Interestingly, in addition to modifying the inflammatory cytokine profile, we also established that PM induces changes in the activity of MnSOD, an enzyme found exclusively in the mitochondrial matrix. Similar to the findings of previous studies indicating that PM also inhibits MnSOD activity in various different tissues, in addition to its effect on muscle tissue, we found that PM significantly reversed the exercise-induced increase in MnSOD activity. Additionally, it has previously been observed that the lung tissues of mice exposed to high concentrations of PM_2.5_ for three months were characterized by a significantly lower GSH/GSSG ratio, while the findings of another study on PM_2.5_-induced lung fibrosis revealed a reduction in the GSH/GSSG ratio and MnSOD activity [49,50]. It is worth noting that the gastrocnemius muscle of both the control and exercise groups of mice examined in the present study were characterized by reductions in the GSH/GSSG ratio, which we speculate could be attributable to the systemic dissemination of inflammatory factors induced by PM in the lungs and airways, as stated in the introduction. However, it is also plausible that PM with extremely small particle size can be deposited directly in muscle tissue. Accordingly, further research is required to determine the precise contributory mechanisms.

Muscle loss and weakness associated with reduced physical activity and exercise are common features of a range of disorders, including diabetes, cancer, kidney failure, and heart failure, and occur as part of the general aging process, a condition referred to as sarcopenia [51]. This catabolic state is associated with marked changes in mitochondrial content, morphology, and function. A deterioration in skeletal muscle functional capacity may involve a reduction in oxidative capacity and resistance to fatigue [52], which may occur because of mitochondrial dysfunction. Mitochondria are the primary energy-producing organelles in cells that support a diverse range of biological processes associated with metabolism, growth, and the regeneration of skeletal muscle [53]. The maladaptive responses linked to malfunctioning mitochondria are attributed to changes in mitochondrial quality control, which includes mitochondrial synthesis (biogenesis), remodeling (dynamics), and degradation (mitophagy) [54,55]. Physical activity contributes to enhancing mitochondrial function by activating mitochondrial biogenesis and mitophagy, which may underlie the beneficial effects of physical activity in the context of several diseases [56]. As anticipated, we discovered that exercise was associated with a significant increase in the mtDNA/nDNA ratio of skeletal muscle. Nevertheless, we found no evidence to indicate that PM inhalation influences mitochondrial biogenesis, which thereby tends to indicate that mitochondrial quality control, which reflects mitochondrial homeostasis, is activated by other factors. Mitophagy has frequently been evaluated by quantifying proteins associated with this process. For example, it has been established that in skeletal muscle, mitophagy is mediated via the activation of Unc-51-like autophagy-activating kinase (ULK1), resulting in the formation of autophagosomes. Additionally, the expression of BNIP3/NIX has been found to trigger the initiation of mitophagy. Subsequently, light chain 3-I (LC3-I) undergoes conversion to a lipid-modified form, light chain 3-II (LC3-II, the phosphatidylethanolamine conjugated form of LC3-I). Additionally, the autophagosomal structural protein, p62/SQSTM1, recruits damaged mitochondria, which leads to the clearance of damaged mitochondria via the degradative activity of LC3. Finally, lysosomal degradative enzymes degrade the mitochondria after several additional processes [57,58]. However, although multiple studies have quantitatively analyzed these proteins to assess mitophagy, it is notably more difficult to characterize the dynamics of mitophagy in vivo by scrutinizing these proteins as a static representation. Consequently, in this study we employed an in vivo model of mitophagy, using mt-Keima mice, which express a pH-dependent fluorescent protein, mt-Keima, thereby enabling a more intuitive and accurate assessment of mitophagy [32]. Using this model, we established that exercise was associated with an increase in in vivo mitophagy, as previously observed in other studies. Moreover, the level was significantly increased by the PM, with values 4–6 times higher than those observed in the EX and PME groups. Accordingly, this in vivo PM-induced increase in mitochondrial removal signaling within skeletal muscles can be considered a novel finding of the present study. Nevertheless, the systemic effects of PM remain unclear, and it is yet to be established whether localized inflammatory and oxidative stresses are promoted directly by the deposition of PM in tissues. In addition, further studies are needed to determine whether PM molecules mediate regulation of the specific proteins involved in mitophagy.

Using permeabilized red gastrocnemius muscle fibers to assess mitochondrial function, we detected increases in both O_2_ consumption and respiratory control ratio for co mplex I + II in response to exercise, whereas levels tended to decline in mice exposed to PM treatment, albeit non-significantly. Contrastingly, exposure to PM induced a significant increase in the production of H_2_O_2_, with the most pronounced effects being detected in response to the inhalation of PM during exercise. Collectively, these findings indicate that although PM triggers a marked increase in mitochondrial ROS production, this does not appear to significantly impair mitochondrial respiration. This outcome would imply that the implicated reactions are controlled dose-dependently by pollutants, such as PM, which should be verified by further experiments assessing different PM exposure concentrations, and durations and intensities of exercise. As an initial investigation into the impact of PM hyper-inhalation on skeletal muscles during exercise, the findings of this study certainly merit additional validation regarding diseases affecting muscle. Models of sarcopenia, an age-related degeneration of skeletal muscle, can also be used. Furthermore, conducting additional studies to examine the effects of different types and intensities of exercise could contribute to the development of appropriate exercise programs to counteract the potentially detrimental effects of exposure to ambient air pollution, particularly that attributable to PM. Additionally, it remains unclear whether intramuscular adhesion of PM or circulatory effects of inflammatory responses in the respiratory system, such as the lungs, are responsible. Any recommendations in this regard should, nevertheless, consider the tradeoff between the benefits of physical activity and the adverse effects of PM inhalation.

## 5. Conclusions

PM adheres to the lung bronchi and alveoli, causing localized inflammation. Given its small particle sizes, PM_2.5_ can circulate systemically, and has potentially adverse effects on the heart, blood vessels, liver, and kidneys. Our findings in this study reveal that inhaling PM also exacerbates inflammation and oxidative stress in skeletal muscle. We also observed mitochondrial oxidative stress, which is comparable to excessive inhalation of PM during exercise (Figure 6). Further studies should focus on analyzing the effects of exposure to different concentrations of PM, in conjunction with different intensities and durations of exercise, which could contribute to guiding exercise schedules for PM-polluted environments.

## Figures and Tables

**Figure 1 antioxidants-13-00113-f001:**
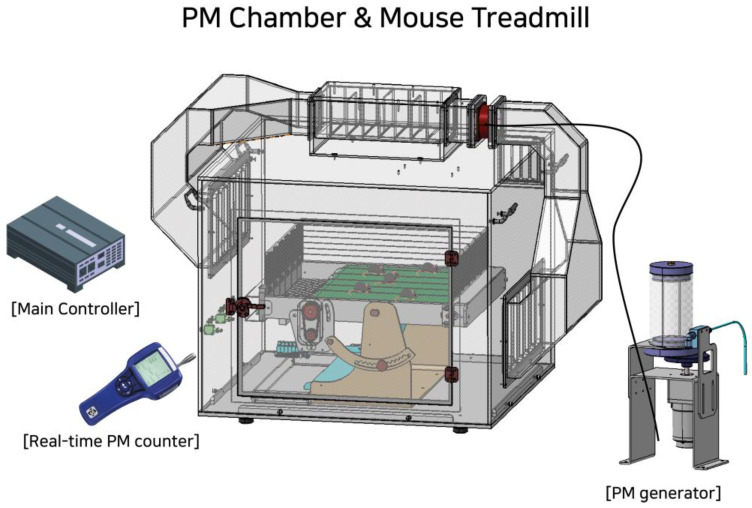
A schematic diagram of the particulate matter (PM) chamber used in this study. The PM generator introduces fine dust into the chamber, the concentration of which is continuously monitored in real-time using an AeroTrak Handheld Particle Counter 9303 (TSI, Shoreview, MN, USA). The PM chamber and its main controller include a mouse treadmill with adjustable speed and duration.

**Figure 2 antioxidants-13-00113-f002:**
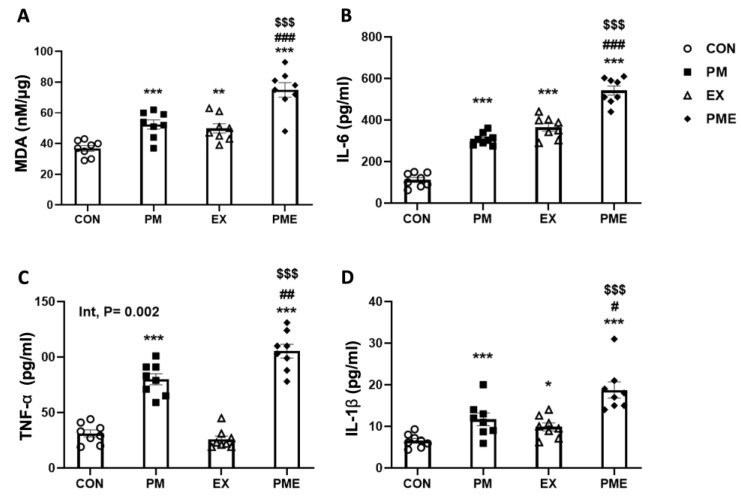
Inflammatory response levels in the gastrocnemius muscle of C57BL/6 mice exposed to particulate matter during exercise. (**A**) MDA (malondialdehyde), (**B**) IL-6, (**C**) TNFα, and (**D**) IL-1β. Control (CON, n = 8), particulate matter exposure (PM, n = 8), exercise (EX, n = 8), and particulate matter exposure + exercise (PME, n = 8). Data are expressed as the means ± SEM. Statistical significance is assigned as * *p* < 0.05, ** *p* < 0.01, *** *p* < 0.001 vs. CON, ^#^ *p* < 0.05, ^##^ *p* < 0.01, ^###^ *p* < 0.001 vs. PM and ^$$$^ *p* < 0.001 vs. EX. Two-way ANOVA results were shown with *p* values. Int, interaction.

**Figure 3 antioxidants-13-00113-f003:**
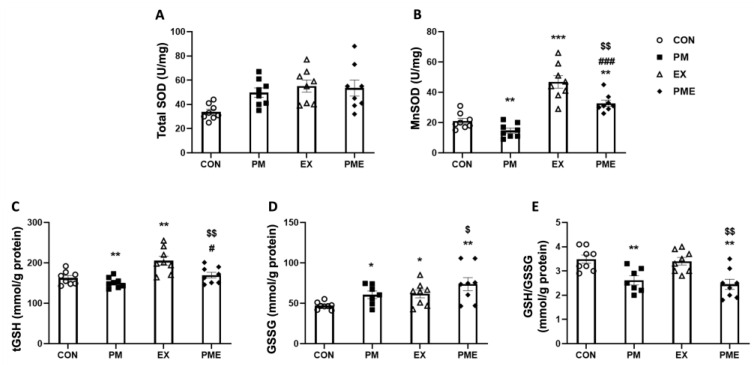
Redox status in the gastrocnemius skeletal muscle of C57BL/6 mice exposed to particulate matter during exercise. (**A**) Total superoxide dismutase (SOD), (**B**) manganese superoxide dismutase (MnSOD), (**C**) total glutathione (tGSH), (**D**) glutathione disulfide (oxidized glutathione, GSSG), and (**E**) GSH/GSSG ratio. Control (CON, n = 8), particulate matter exposure (PM, n = 8), exercise (EX, n = 8), and particulate matter exposure + exercise (PME, n = 8). Data are expressed as the means ± SEM. Statistical significance is defined as * *p* < 0.05, ** *p* < 0.01, *** *p* < 0.001 vs. CON, ^#^ *p* < 0.05, ^###^ *p* < 0.001 vs. PM and ^$^ *p* < 0.05, ^$$^ *p* < 0.01 vs. EX.

**Figure 4 antioxidants-13-00113-f004:**
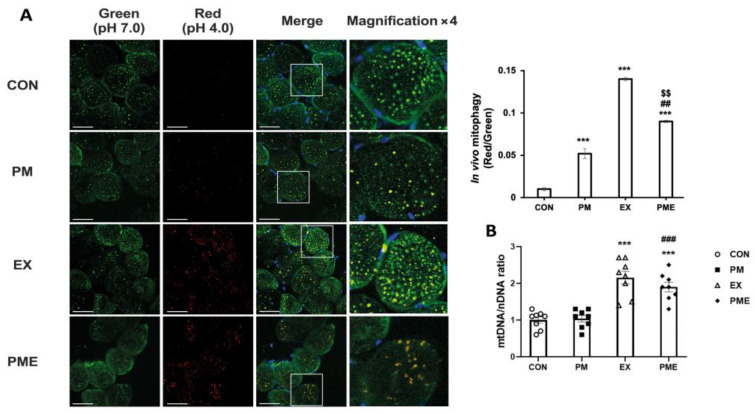
In vivo mitophagy levels in the skeletal muscles of mt-Keima mice were modified by particulate matter exposure and exercise. (**A**) Representative confocal images showing superimposed red/green signals in the skeletal muscle of mt-Keima mouse. The yellow signal Indicates merged red and green fluorescence. The white square indicates area shown magnified in each inset (4x magnification). Control (CON, n = 4), particulate matter exposure (PM, n = 4), exercise (EX, n = 4), and particulate matter exposure + exercise (PME, n = 4). (**B**) Level of mitochondrial biogenesis (mtDNA to nDNA ratio). Control (CON, n = 8), particulate matter exposure (PM, n = 8), exercise (EX, n = 8), and particulate matter exposure + exercise (PME, n = 8). Data are presented as the mean ± SEM. Statistical significance is assigned as *** *p* < 0.001 vs. CON, ^##^ *p* < 0.01 and ^###^ *p* < 0.001 vs. PM, and ^$$^ *p* < 0.01 vs. EX. Scale bar, 50 μm.

**Figure 5 antioxidants-13-00113-f005:**
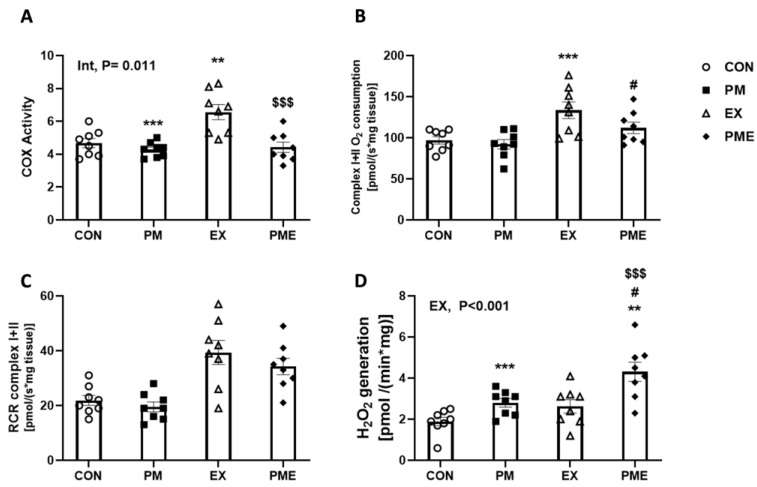
Mitochondrial respiratory function in the muscles of C57BL/6 mice exposed to particulate matter during exercise. (**A**) COX activity assay, (**B**) O_2_ consumption, (**C**) respiratory control ratio (RCR), (**D**) basal rate of hydrogen peroxide (H_2_O_2_) generation. Control (CON, n = 8), particulate matter exposure (PM, n = 8), exercise (EX, n = 8), and particulate matter exposure + exercise (PME, n = 8). Data are presented as the means ± SEM. Statistical significance is assigned as ** *p* < 0.01, *** *p* < 0.001 vs. CON, ^#^ *p* < 0.05 vs. PM and ^$$$^ *p* < 0.05 vs. EX. Two-way ANOVA results were shown with *p* values. Int, interaction; EX, exercise.

**Figure 6 antioxidants-13-00113-f006:**
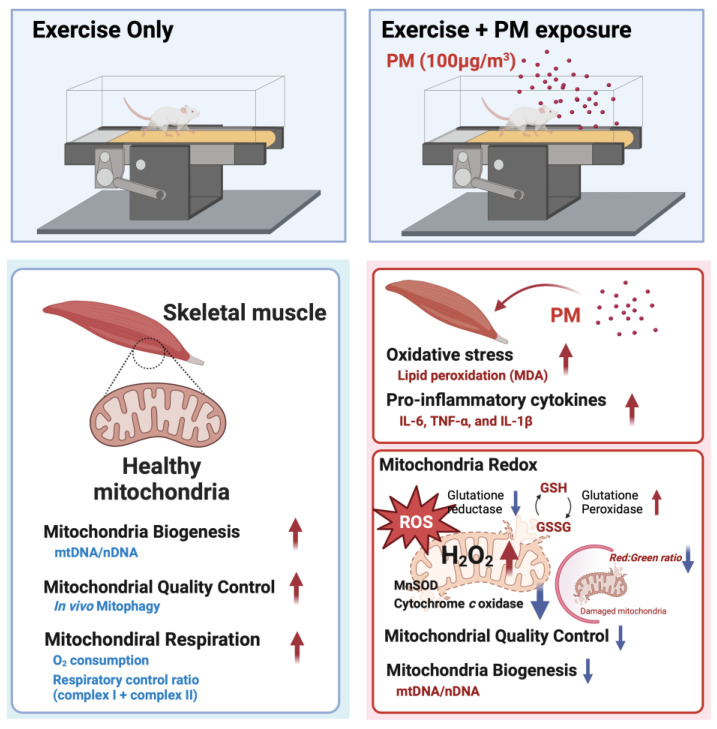
A graphical summary depicting the impact of particulate matter (PM) and exercise on mouse skeletal muscle and mitochondrial function. The exercise group exhibited an increase in mitochondrial biogenesis (mtDNA/nDNA), mitophagy, and mitochondrial respiration. PM inhalation during treadmill exercise leads to increased oxidative stress and elevated levels of proinflammatory cytokines, as evaluated based on analyses of malondialdehyde (MDA) and interleukin-6 (IL-6), tumor necrosis factor-alpha (TNF-α), and interleukin-1 beta (IL-1β). The ratio of reduced (GSH) to oxidized (GSSG) glutathione levels is reduced in response to PM exposure, and mitochondrial biogenesis and mitophagy are reduced in the muscles of mice exposed to PM during exercise.

## Data Availability

The data obtained in this study are available upon reasonable request to the corresponding author.

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
