# Peer review of "Effects of Particulate Matter Inhalation during Exercise on Oxidative Stress and Mitochondrial Function in Mouse Skeletal Muscle"

_antioxidants, 2024, doi:10.3390/antiox13010113_

Round 1

Reviewer 1 Report

Comments and Suggestions for Authors

The main goal of this article is to assess the effect of PM inhaled during exercise on mouse skeletal muscle.  

The aim of the study is interesting and the technique used to treat the animals in the study is appropriate. However, the results of the study  are not presented and discussed appropiately.  Not all significances between histograms are reported and there is not always corrispondence what is stated ion discussion and what is observed in the figures.

In detail

pag. 2 line 50 Please remove "facilitating"

page 2 line 61 The interplay between exercise and oxidative stress and ROS production deserve a broader discussion in order to clarify  the exercise modes that are accompanied by increased oxidative stress and which are not

pag.2 paragraph 2.1 The Authors could explain why only female mice are used for the experiments

pag.3 paragraph 2.3 From "These mice (line 105) to [25,26] (line 109) : it could be better to move the description of the mice in the paragraph 2.1 Experimental animals. Morover the charateristic of the mice must be explained in more details. It looks like the mice turn red or green.

In the Figure 2, 3, 4 and 5  The Authors should add the significances between CON and EX,  between EX and PME and between PME and PM 

line 257 "Similarly the RCR level was not different between teh CON and PM" .  In the figure 5 it seems to me that PM was significantly different from CON ( *** p < 0.001 vs. CON)

line 364 "The PM intervention further augmented this level, exceeding that observed in the EX and PME groups" In the figure 4 It seems to me that PM intervention doesn't augmented the level observed in EX and PME

line 374 "O2 consumption and respiratory control ratio (RCR) for complex I + II were both increased by exercise and tended to decrease with PM treatment" . In the figure 5 it seems to me that RCR doesn't increase by exercise and doesn't decrease with PM

line 384 "this study presents preliminary findings on the creation of appropriate exercise recommendations in response to ambient air pollution" On the basis of obtained results  I dont think we can create exercise raccomandation

Author Response

1) Firstly, page. 2 line 50 Please remove "facilitating"

  • We removed the “facilitating”.

2) Page 2 line 61 The interplay between exercise and oxidative stress and ROS production deserves a broader discussion in order to clarify the exercise modes that are accompanied by increased oxidative stress and which are not

  • Thank you for pointing this out. Therefore, I describe more detail of exercise interplay between oxidative stress and ROS.

(Page 2. line 66-76) Under normal physiological conditions, ROS play a key role in various cellular activities, including cellular energy metabolism, signal transduction, and the regulation of gene expression. However, when produced in excess, they can cause damage to cellular biomolecules, including lipids, proteins, and nucleic acids, thereby promoting cellular aging and eventually cell death [21]. High-intensity acute exercise, accompanied by a rapid increase in oxygen consumption, promotes excessive ROS production, leading to an imbalance in the oxidative–antioxidative homeostasis of cells [22]. However, in skeletal muscle adapted through long-term endurance training, the fibers undergo various structural adaptations and an increase in antioxidant enzyme activity, enhancing the capacity to adapt to high-intensity acute exercises and reducing oxidative stress levels [23,24].

3) Page.2 paragraph 2.1 The Authors could explain why only female mice are used for the experiments.

  • As you mentioned, we have added the rationale for using female mouse as follows.

(Page 2. line 92-93) To eliminate the potential effects of testosterone on the accumulation of ROS and mitochondria-mediated cell death, [29,30]

4) Page.3 paragraph 2.3 From "These mice (line 105) to [25,26] (line 109) : it could be better to move the description of the mice in the paragraph 2.1 Experimental animals. Moreover, the characteristic of the mice must be explained in more details. It looks like the mice turn red or green.

  • We have moved the sentence to better reflect what you said and added more detail about mt-Keima mice.

(Page 3. Line 98-102) These mice harbor a mitochondria-targeting sequence derived from COX VIII that binds to the pH-dependent fluorescent protein Keima, thereby facilitating the detection of mitophagy. This model reflects the physiological status of mitochondria based on a dual-fluorescence probe, with a green fluorescence being indicative of normal conditions (pH 8.0) and red fluorescence signifying acidic lysosomal conditions (pH 4.5) [31,32].

5) In the Figure 2, 3, 4 and 5 The Authors should add the significances between CON and EX, between EX and PME and between PME and PM.

  • Thank you for bringing this to our attention. We compared all groups and indicated significant differences.

6) line 257 "Similarly the RCR level was not different between the CON and PM".  In the figure 5 it seems to me that PM was significantly different from CON ( *** p < 0.001 vs. CON).

  • We rechecked the raw data and found no significant difference between PM and CON.

7) line 364 "The PM intervention further augmented this level, exceeding that observed in the EX and PME groups" In the figure 4 It seems to me that PM intervention doesn't augmented the level observed in EX and PME.

  • We've made the following changes to provide more accurate information.

(Page 10. Line 391-392) the level was significantly increased by the PM, with values 4-6 times higher than those observed in the EX and PME groups.

8) line 374 "O2 consumption and respiratory control ratio (RCR) for complex I + II were both increased by exercise and tended to decrease with PM treatment". In the figure 5 it seems to me that RCR doesn't increase by exercise and doesn't decrease with PM

We apologize for the mistake in switching the groups while plotting the graph based on the raw data. We have reworked the graph to accurately reflect the raw data. Thank you for bringing this to our attention.

9) line 384 "this study presents preliminary findings on the creation of appropriate exercise recommendations in response to ambient air pollution" On the basis of obtained results I don’t think we can create exercise recommendation

  • Thanks for the feedback. We have made the following changes to eliminate any room for exaggeration and added study limitations.

(Page 10. Line 412-420) Furthermore, conducting additional studies to examine the effects of different types and intensities of exercise could contribute to the development of appropriate exercise programs to counteract the potentially detrimental effects of exposure to ambient air pollution, particularly that attributable to PM. Additionally, it remains unclear whether intramuscular adhesion of PM or circulatory effects of inflammatory responses in the respiratory system, such as the lungs, are responsible. Any recommendations in this regard should, nevertheless, consider the tradeoff between the benefits of physical activity and the adverse effects of PM inhalation.

Reviewer 2 Report

Comments and Suggestions for Authors

In the current study, the authors investigated the effects of particulate matter on mouse skeletal muscle. This is an interesting study, nevertheless, it required some improvements.

Questions/ suggestions/ limitations of the study.

Line 89: Please provide the date of the approval.

Line 93: Please describe why “Urban Particulate Matter” samples were used for this study. What size particles it contains?

Figures 2, 3, 4 and 5. Indications of the statistical significance are unclear. For example, in Figure 2B EX and PME samples are significantly higher than the CON sample, but there is no indication of the statistical significance. Please modify. In Figure 3B EX sample is significantly higher than the PM sample, but there is no indication of the statistical significance.

Line 219: Figure 4B is mentioned before 4A. Please modify.

Figure 4A is very dark and it is impossible to see the staining there. Please provide better staining and views on both low-magnification and high-magnification of muscle samples.

Line 285: “Multiple studies have shown elevated oxidative stress in the skeletal muscle following exercise due to the rise of ROS in the form of free radicals like superoxide anion radicals (O2•−), hydroxyl radicals (·OH), hydroperoxyl radicals (HOO.), singlet oxygen (1O2), and free nitrogen radicals [33].”

This is repeated in Line 292.

“Numerous studies have shown that various types of exercise lead to increased oxidative stress in skeletal muscle due to the rise in free radicals such as superoxide anion radicals (O2•−), hydroxyl radicals (·OH), hydroperoxyl radicals (HOO.), singlet oxygen (1O2) and free nitrogen radicals [33]”.

Lines 352-357: The current study has not evaluated the expression of the proteins listed there. Why are they discussed?

Please provide a discussion on the limitations of this study.

Comments on the Quality of English Language

Minor editing of English language required.

Author Response

1) Line 89: Please provide the date of the approval.

  • Thank you for bringing this to our attention. We have added the date of the approval.

(Page 3. line 106-107) IACUC: approval number INHA 190211-616, approval date: 2019.02.11; TJUS: approval number TJUS 2022-021, approval date: 2022.02.

2) Line 93: Please describe why “Urban Particulate Matter” samples were used for this study. What size particles it contains?

  • Thank you for bringing this to our attention. The reason for using the “Urban Particulate Matter” sample in this study is to confirm the negative effects of PM more accurately by making it similar to an actual PM environment. Particle sizes are presented in the table below.

Mean Particle Diameter, d (0.5) 5.85 μm

Particle Diameter, d (0.1) 1.35 µm

Particle Diameter, d (0.9) 30.1 µm

Figure 1. Particle size distribution for SRM 1648a after 10minute sonication in water. Solid line represents the volume in %. See “Particle-Size Information (Table 10)” for additional information.

3) Figures 2, 3, 4 and 5. Indications of the statistical significance are unclear. For example, in Figure 2B EX and PME samples are significantly higher than the CON sample, but there is no indication of the statistical significance. Please modify. In Figure 3B EX sample is significantly higher than the PM sample, but there is no indication of the statistical significance.

  • Thank you for bringing this to our attention. We compared all groups again and indicated significant differences.

4) Line 219: Figure 4B is mentioned before 4A. Please modify.

  • We corrected our mistake by swapping the positions of the paragraphs, as you pointed out.

5) Figure 4A is very dark and it is impossible to see the staining there. Please provide better staining and views on both low-magnification and high-magnification of muscle samples.

  • The resolution of the graph in the Word file was poor, unlike the original image. We have improved it to more accurately reflect the original picture. We included a high-magnification illustration, as shown in the figure.

6) Line 285: “Multiple studies have shown elevated oxidative stress in the skeletal muscle following exercise due to the rise of ROS in the form of free radicals like superoxide anion radicals (O2•−), hydroxyl radicals (·OH), hydroperoxyl radicals (HOO.), singlet oxygen (1O2), and free nitrogen radicals [33].”

 This is repeated in Line 292.

“Numerous studies have shown that various types of exercise lead to increased oxidative stress in skeletal muscle due to the rise in free radicals such as superoxide anion radicals (O2•−), hydroxyl radicals (·OH), hydroperoxyl radicals (HOO.), singlet oxygen (1O2) and free nitrogen radicals [33]”.

  • I apologize for not paying attention. We have deleted one of the redundant paragraphs.

7) Lines 352-357: The current study has not evaluated the expression of the proteins listed there. Why are they discussed?

  • Thank you for your feedback. We aimed to explain the limitations of typical protein quantification and emphasize the importance of in vivo mitophagy. We thought it is worth noting that most studies analyze the proteins listed.

 8) Please provide a discussion on the limitations of this study.

  • Thanks for the feedback. We have added study limitations as belows; We have made the following changes to eliminate any room for exaggeration and added study limitations.

(Page 10. Line 412-420) Furthermore, conducting additional studies to examine the effects of different types and intensities of exercise could contribute to the development of appropriate exercise programs to counteract the potentially detrimental effects of exposure to ambient air pollution, particularly that attributable to PM. Additionally, it remains unclear whether intramuscular adhesion of PM or circulatory effects of inflammatory responses in the respiratory system, such as the lungs, are responsible. Any recommendations in this regard should, nevertheless, consider the tradeoff between the benefits of physical activity and the adverse effects of PM inhalation.

9) Comments on the Quality of English Language: Minor editing of English language required.

  • Minor revisions have been made to the wording of English sentences throughout the manuscript. Thank you for the feedback.

Round 2

Reviewer 1 Report

Comments and Suggestions for Authors

The work has significantly improved. It is now suitable for pubblication

Author Response

Thank you for reviewing our manuscript and providing valuable feedback.

Reviewer 2 Report

Comments and Suggestions for Authors

The authors significantly modified and improved the manuscript. Nevertheless, several questions remain.

1. Please add the description of the Urban Particulate Matter samples to the Methods section of the manuscript.

2. The authors listed Page 10. Line 412-420 as limitations of this study. This text has no mention of the limitations of the study. It states that “conducting additional studies to examine the effects of different types and intensities of exercise could contribute to the development of appropriate exercise programs” and “it remains unclear whether intramuscular adhesion of PM or circulatory effects of inflammatory responses in the respiratory system”. These are not limitations of the study. Please discuss the limitations of your study design, results, or results interpretations.

For example, “The design limitations of the current study do not allow to distinguish whether intramuscular adhesion of PM or circulatory consequences of inflammatory responses to the PM effects in the respiratory system, such as the lungs, are responsible for the skeletal muscle outcomes evaluated here”.

3. With the better-quality images in Figure 4, it is obvious that fibers in PME have much less distinctive green mitochondria than in EX. Please explain what the reason for this is. Could it be that the image for EX shows oxidative fibers and the image for PME shows glycolytic? Could it be that the mitochondria in PME are destroyed and therefore there are fewer intact mitochondria that are stained? Is this the reason for the decreased COX activity in PME samples in Figure 5? This requires an explanation in the text of the manuscript.

Author Response

(The authors gave the same response as above.)
